# Microbial-Induced Carbonate Precipitation Improves Physical and Structural Properties of Nanjing Ancient City Walls

**DOI:** 10.3390/ma14195665

**Published:** 2021-09-29

**Authors:** Baogang Mu, Zheyi Gui, Fei Lu, Evangelos Petropoulos, Yongjie Yu

**Affiliations:** 1School of Civil Engineering, Southeast University, Nanjing 211189, China; guizheyi@163.com (Z.G.); lufei@seu.edu.cn (F.L.); 2Key Laboratory of Urban and Architectural Heritage Conservation of Ministry of Education, Southeast University, Nanjing 210096, China; 3School of Engineering, Newcastle University, Newcastle upon Tyne NE1 7RU, UK; vagpetrop@gmail.com; 4School of Applied Meteorology, Nanjing University of Information Science & Technology, Nanjing 210044, China

**Keywords:** MICP, X-ray micro-CT, restoration of cultural relics, ancient wall bricks

## Abstract

The preservation and restoration of heritage sites have always been of key focus in the field of cultural relics. Current restoration methods mainly involve physical or chemical techniques, which are in many cases intrusive, destructive, and irreversible. Hereby, we introduce a novel biological strategy (microbial-induced carbonate precipitation (MICP)) to repair natural and simulated surface cracks on six hundred years’ old wall bricks (part of the Nanjing City Min Dynasty ancient wall, China). X-ray micro computed tomography (X-ray micro-CT) was employed to non-destructively visualize the internal structure of the MICP-treated brick cubes. The results showed that MICP can effectively repair both natural and simulated cracks present on the brick’s surface. The compressive strength of the MICP-treated brick cubes was significantly higher than that of the untreated control cubes (33.56 ± 9.07 vs. 19.00 ± 1.98 kN, respectively). MICP significantly increased the softening coefficient and decreased the water absorption rate (*p* < 0.05), indicating that the water resistance of the wall bricks can be improved after treatment. The 3D images from X-ray micro-CT, a method that could non-destructively assess the internals of such cultural structures, showed that MICP can effectively repair ancient relics, promoting durability and limiting degradation without affecting the structure. X-ray diffraction analyses showed that MICP generates the same calcite form as that of original bricks, indicating that MICP filler is compatible with the ancient city wall brick. These findings are in line with the concept of contemporary heritage preservation.

## 1. Introduction

As per the evolution of civil architecture, numerous traditional underground cultural relics (mausoleums) and magnificent above-ground ancient buildings were constructed of masonry with sintered clay bricks. Among them is the Nanjing City Wall, an outstanding representation of above-ground architecture, one of the longest, largest, and most authentic ancient city walls worldwide. Nanjing City Wall, built in 1358 AD (the first year of Hongwu in the Ming Dynasty), has a history of more than 600 years [1]. Apart from the prominent risk of systemic load-bearing capacity, the most common problem of the Nanjing City Wall is the persistent deterioration of the main brick material and the development of cracks on its surface driven by weathering. A local survey on 25 km of Nanjing’s city wall showed that 60% of the bricks have megascopic cracks on their surface. Inevitably, the persistent deterioration will eventually lead to a reduction in load-bearing capacity increasing the risk of failure. Therefore, the city wall bricks urgently require the establishment of means for weathering mitigation or other deterioration prevention measures to eliminate the likelihood of failure.

At present, grouting with a mortar-type binder is a commonly used engineering strategy to reinforce and seal cracks as such (Table 1). However, sealing renders structure post-assessment impossible, while qualitatively, the stiffness and strength of the seal rarely match that of the original reinforced structure. Such physicochemical remedial strategies may also affect the original appearance and the overall aesthetics of such structures [2]. Nonetheless, the high alkalinity of mortar-type binder would increase the risk of efflorescence of the original masonry structure. A plethora of cracks may result in a “colorful” disharmony on the wall, which may, overall, greatly damage the historical value of cultural relics. This renders service providers of such heritage sites hesitant towards using traditional sealing methods.

Microbial-induced carbonate precipitation (MICP) is a biochemical reaction involving the bacterial hydrolysis of urea into ammonium and carbonate ions in alkaline environments [5,6]. In the presence of calcium ions, CaCl_2_, carbonate radicals react to produce calcium carbonate precipitation [7]. The chemical reactions of this process are as follows.
$${\left({\mathrm{NH}}_{2}\right)}_{2}\mathrm{CO}+3H_{2}O\to 2{\mathrm{NH}}_{4}^{+}+{\mathrm{HCHO}}_{3}^{-}+{\mathrm{OH}}^{-}$$
$${\mathrm{HCHO}}_{3}^{-}+{\mathrm{OH}}^{-}+{\mathrm{Ca}}^{2+}\to {\mathrm{CaCO}}_{3}$$

Microorganisms generally carry a great number of negative functional groups (such as hydroxyl, amine, amide, carboxyl, etc.) on their cell wall surface. They can adsorb Ca^2+^ from the solution as crystallization nuclei, producing microbial-induced calcite bridges on the adjacent particles to form cemented bio-calcareous rock [8]. Inspired by the natural ability of MICP, researchers have used this technique as a promising self-healing technology for concrete crack remediation and improvement of ash bricks [9,10]. Therefore, MICP could also be a biological control tool against the cracks of city wall bricks as well as for other ancient remnants subjected to deterioration from uncontrolled cracking.

As the opaque nature precludes in situ visualization of the internals of cultural relics, it is difficult to directly visualize their restoration degree. At present, almost all reports related to MICP revealed the restoration effect via scanning electron micrographs (SEM) and confirmed it by X-ray diffraction (XRD). However, the microscopic area observed/covered by SEM is too small to evaluate the overall sealing degree of a macroscopic crack. Thus, there is an urgent need to establish a method to view in three-dimensional (3D) imaging the internals of the cultural relics using non-invasive and non-destructive techniques.

In recent years, X-ray micro-CT (micro-CT) has been used in a wide range of disciplines to visualize the 3D structure of different materials, including biology and earth sciences, etc. [11,12]. The development of high-resolution scanning techniques has also been used to assess micro-structures such as the visualization of soil pores from images [13], the characterization of aggregation processes [14], and the stimulation of hydraulic stresses [15]. Compared to SEM and XRD, micro-CT imaging has two advantages: (i) detecting 3D inner spaces of the samples, and (ii) the ability to do this without having to destroy or even damage the sample. Therefore, we hypothesize that it can be a representative tool to study the restoration effect of relics subjected to MICP treatment.

The present study aims to: (i) present MICP as a sustainable method for sealing the cracks of the ancient city bricks and enhance their strength and durability; (ii) evaluate the restoration effectiveness of MICP by visualizing and quantifying the internal structure of the bricks using innovative high-resolution micro-CT.

## 2. Materials and Methods

### 2.1. Cubical Samples from the Ancient City Wall Brick

The ancient city wall brick samples were collected from one of the original gates of the Nanjing City Wall, the Jiefang Gate, which is situated at the northern part of the wall. The ancient city wall was built during China’s early Ming dynasty [1] in the mid-to-late 14th century. The wall project has been under construction for 28 years and millions of workers have been involved in these construction works. The wall was comprised of four parts: (i) the palace walls, (ii) the imperial city walls, (iii) the inner-city walls, and (iv) the outer city walls. The inner-city wall has 13 gates and is 35.3 km long. The outer city wall has 18 gates with a length of 60 km. Nanjing City has a mean annual temperature of 15.8 °C, with a maximum mean temperature of 36.8 °C in summer and a minimum mean temperature of −5.5 °C in winter. Diurnal temperature variation is about 8.4 °C in summer and about 5.9 °C in winter. The mean annual humidity of Nanjing is 74.9%. The traffic flow in inner Nanjing City is about 132 K vehicles per day. The acid rain frequency in the past 5 years is about 40% with a mean annual pH value of 5.09. Due to natural erosion, human activities, and other historical events, only 25.1 km and 45 km of inner and outer walls, respectively, remain standing. The existing wall bricks are in urgent need of restoration and protection (Figure 1A,B). The tested bricks were obtained from a partially damaged area of the ancient city wall. From the collected city wall bricks, cubes with a size of 5 cm × 5 cm × 5 cm were abstracted using an electric saw. The cubes collected were rich in obvious natural cracks on their surface (Figure 1C) and were ideal for the scope of our experiment. For those cubes without evident cracks, an electric circular saw was employed to simulate natural cracks on the surface of the brick cubes (Figure 1D). Finally, three kinds of brick cubes were obtained: (i) brick cubes with obvious natural cracks, (ii) brick cubes with simulated cracks, and (iii) original cubes without any observable cracks.

### 2.2. Microbial Induced Carbonate Precipitation Method

*Sporosarcina pasteurii* (CGMCC 1.3687), an ureolytic bacterium purchased from China General Microbiological Culture Collection Center (Beijing, China), was employed to mediate MICP remediation. The culture medium contained 5 g/L peptone, 3 g/L yeast extract, and 20 g/L urea at a pH of 8, to which 1.5% agar was added to obtain the solid medium. The medium was then autoclaved to promote sterility; an exception was the urea that was filter-sterilized to prevent decomposition from heat. *S. pasteurii* was cultivated at 30 °C under aerobic conditions subjected to shaking at 180 rpm. Until the end of the exponential phase, the media was centrifuged to obtain a high concentration of bacterial cells and then stored at 4 °C until further use.

Each group of the prepared cubes was first placed into 1 M CaCl_2_ solution at a temperature of 28 °C for 4 h. The concentrated bacterial cells after centrifugation were suspended in the culture medium at a cell concentration of 10^8^ cell/mL and then mixed with silica powder (SiO_2_) at a ratio of 5:1 (*v*/*w*). Silica slurry containing both active/inactive cells was applied to the narrow cracks of the cubes and compacted using a flat spade. CaCl_2_ solution (1 M) was added to the silica slurry homogeneously during the crack filling multiple times until a uniform seal was achieved. Finally, the surface of the cubes was smoothed using a spatula. All cubes were air-dried at room temperature for four weeks.

### 2.3. Compression Strength and Water-Resistance Test

The compression testing was performed via an automatic compression testing machine (ASTM C39, Wuxi Serve Real Technology Co., Ltd., Wuxi, China) according to the standard test methods for the assessment of the mechanical properties of ordinary concrete (GB/T50081-2002) [16]. The statistical significance of the differences between the samples was performed by Tukey’s HSD post hoc test using SPSS 16.0 (SPSS Inc., Chicago, IL, USA).

To assess the water resistance of the brick cubes after the MICP method, the softening coefficient was calculated according to the Chinese national standard GB/T 4111-2013 [17]. In detail, the brick cubes were dried at 105 °C for 24 h and the compressive strength was measured when the sample was in a dry state. For compressive strength in a water-saturated state, the brick cubes were immersed into sterile water (20 ± 1 °C) for 24 h at room temperature. The softening coefficient (*K*) was calculated by the following formula: (1)$$K=\frac{I_{1}}{I_{0}}$$
where *I*_1_ is the compressive strength of a brick cube sample saturated with water; *I*_0_ is the compressive strength of the brick cube sample in the dry state.

The water absorption test was performed according to BSEN 12087:2013. The brick cubes were first placed into a 105 °C oven and remained there for over 24 h, the dry mass was then recorded. Afterwards the cubes were immersed into sterile water (20 ± 1 °C) for 20 min, 40 min, 60 min, 120 min, and 24 h, at these time intervals they were taken out of the water for mass measurements. The water absorption ratio (*R*_n_) was calculated by the following formula: (2)$$R_{n}=\frac{m_{n}-m_{0}}{m_{0}}$$
where *m*_n_ is the mass of brick cubes immersed in sterile water for a specific time; *m*_0_ is the mass of dry brick cubes.

### 2.4. X-ray Scanning Visualization

X-ray diffraction (XRD) analysis was performed on an Ultima IV multipurpose X-ray diffraction system (Rigaku Corporation, Tokyo, Japan). X-ray fluorescence (XRF) was carried out on an HS-XRF system (Beijing Ancoren Tech Co., LTD., Beijing, China). The MICP treated sample was scanned using μCT (GE, Sensing and Inspection Technologies, GmbH, Wunstorf, Germany), visualization took place at the Institute of Soil Science, Chinese Academy of Sciences. Each cube was mounted on a rotary stage and rotated from 0° to 360°, 1800 absorption radiographs were acquired from different angles during rotation. Reconstruction of the visual slices was conducted with the Datos × 2.0 software (GE, Sensing and Inspection Technologies, GmbH, Wunstorf, Germany), using the filtered back-projection algorithm. A total of 2000 slices, each with a size 2000 × 2000 pixel were reconstructed. The resulting voxel size was 30 × 30 × 30 μm^3^, with voxel values ranging from 0 to 255 corresponding to the attenuation coefficient.

The images of the MICP-treated brick cube were processed using the ImageJ software (Version 1.50). Since the sample has a regular cubic shape, the region of interest (ROI) was cropped from the samples with negligible loss due to the surface roughness. The resulting size of the imaged cube was 43.92 × 45.60 × 44.85 mm^3^. Images were de-noised using the “median filter” (radius = 2 pixels) in 2D. The repaired sample included three phases, i.e., the matrix, pores, and fillings, each exhibiting different grayscale values. A two-step segmentation strategy was used to separate the three phases. Firstly, the pores and fillings were separated from the matrix. Secondly, the fillings were separated from the pores. The threshold values for the segmentation were determined by visual inspection based on the grayscale histograms. The volume fractions of the pores and fillings were determined as the total volume of the corresponding voxels divided by the volume of ROI.

## 3. Results

### 3.1. Effectiveness of MICP on Megascopic Cracks

For each of the brick cubes with natural cracks, all six cube faces were MICP treated (Figure 2A). For the cubes with simulated cracks, only the obvious cracks made by the circular saw were MICP treated (Figure 2B). The filled material from both natural and stimulated cracks of the city wall brick cubes was not broken or fallen off throughout experimentation (Figure 2). Visually, all megascopic cracks were firmly repaired by MICP treatment for the brick cubes with natural cracks (Figure 2A). The targeted stimulated cracks also appeared to have a satisfactory filling (Figure 2B).

### 3.2. Mechanical Properties of the City Wall Bricks after MICP Treatment

To check whether MICP improved the engineering properties of the brick cubes with stimulated cracks, an automatic compression machine was employed to carry out a compressive strength test (Figure 3). The results showed that MICP effectively improved the mechanical strength of the brick cubes with simulated cracks after remediation (Table 2). In detail, the failure load for the cubes subjected to MICP remediation was 33.56 ± 9.07 kN—a value close to the maximum load of the original cubes without any observed cracks (40.02 ± 3.84 kN). The bricks subjected to MICP remediation had a significantly higher failure load than those cracked and untreated (*p* < 0.05). The compression strength of the MICP treated cubes was 17.33 ± 4.69 MPa, twice as high as that of the cracked/untreated samples (9.84 ± 1.02 MPa). The compression strength of the original un-cracked cubes (20.67 MP) was higher than that of no MICP remediation cubes (9.84 MP). No significant difference between the compressive strength between the MICP remediated and the original un-cracked cubes (*p* > 0.05) was observed. This indicates that simulated cracks significantly decreased both the failure load and compression strength, while MICP treatment could increase both structural parameters.

### 3.3. Effectiveness of MICP on Water Resistance

The stability of the city wall structure is strongly influenced by the water content and the resistance ability of the wall bricks. Therefore, water resistance is a vital property for the remediated bricks to prevent them from strength loss. Figure 4 illustrates the softening coefficient of the original, the MICP-treated, and the original cubes. It is observed that cracked bricks had a lower softening coefficient (0.78 ± 0.05) than that of the original cubes (0.84 ± 0.03). MICP treatment significantly increased softening coefficient compared to the original cubes (*p* < 0.05), suggesting that MICP treatment could improve water resistance of the wall bricks.

The MICP-treated and original cubes were used to study the changes in the water absorption ratio versus immersion time. It could be observed that the water absorption ratio of the untreated original cubes increased the first 40 min and maintained a plateau at a peak value of 11.1%; the water absorption ratio of the MICP-treated brick cubes increased with submergence time (from 20 min to 24 h), however, it remained lower than that from the original samples (Figure 5).

The cubical samples after MICP treatment were examined under microscopy to evaluate the water resistance mechanism (Figure 6). It was observed that the pores on the surface of the cubes were ‘clogged’ with organic and inorganic particles. From the 20× and 50× magnifications (upper part of Figure 6), natural cracks after MICP treatment appeared fully filled. In the higher magnifications, 100× and 200× (lower part of Figure 6), the filling particles appear bonded together forming a film on the surface of the brick cubes.

### 3.4. Effectiveness of MICP on Natural Cracks Revealed by X-ray Micro-CT

The X-ray micro-CT was used to further assess the state of the cracks after MICP treatment (Figure 7). The scanning and image processing allowed for both 2D and 3D throughout visualizations of the pore structures, hence the volume of internal pores could be assessed. In this study, all pores with a volume larger than 30 × 30 × 30 μm^3^ were taken into consideration. The total pores’ volume of the tested brick cube was 5.31 × 103 mm^3^, occupying 5.91% of the total cube (43.92 × 45.60 × 44.85 mm^3^). The natural cracks volume was 4.45 × 103 mm^3^, occupying 83.7% of the pores in the brick cube (Figure 7A). This highlights that the majority of the ‘gaps’ account as natural cracks. After treatment, the MICP particles filled the natural cracks, forming a filler with a volume of 9.25 × 10^2^ mm^3^ (Figure 7B). The volume of MICP filler occupied 17.3% of the total pores and 20.8% of the natural cracks. After MICP treatment, the white-colored precipitants were established on both the surface as well as the internal of the cubes. The white precipitants were composed by solidified silica slurry cemented by produced CaCO_3_. The distributions of the external surfaces of the MICP-treated cubes are shown on Figure 7C. The light grey represents the matrix of the cubes while the dark grey depicts the filled pores on the external surface. Visually, using the external surface as reference, it can be observed that after MICP the filling of the pores was complete and the overall surface appeared smooth.

### 3.5. X-ray Diffraction (XRD) and X-ray Fluorescence (XRF) Analyses

The Mineralogy of the ancient city wall bricks and the MICP filler were assessed by XRD analysis, the results are presented in Figure 8. It was shown that the ancient brick is dominated by a quartz phase (ca. 80%), followed by a hercynite and a feldspar one, small quantities of calcite were also detected (less than 2%). The dominant XRD peaks in MCIP filler displayed between 16–26°, which is identified as the amorphous phase. The other two peaks from the MICP filler were identified as calcite (approximately 2%). Table 3 shows the results of the XRF analysis on the ancient brick and the MICP filler. These findings show that the metal oxides in ancient brick included silica (SiO_2_), calcium oxide (CaO), alumina (Al_2_O_3_), and iron oxide (Fe_2_O_3_). The predominant component in the ancient brick and MICP filler is SiO_2_. The ratio of CaO is 3.25% and 4.31% in the ancient brick the MICP filler, respectively. MICP filler was also found to have a greater ‘other element’ ratio (21.35%) compared to the ancient brick (12.37%).

## 4. Discussion

Ancient architecture includes structures built by masonry with sintered clay bricks. These materials are exposed in ambient and dusty environments, continuously subjected to wind erosion, plurosion, and human activities, phenomena that slowly wither their properties. Therefore, remediation of the surface cracks, and not only, that appear on such ancient bricks is crucial to ensure the preservation of masonry relics. In this study, we applied a microbial-induced carbonate precipitation method to remediate natural and simulated cracks on city wall bricks and evaluated how this biological technique improves the structural properties of these materials.

### 4.1. MICP Improved the Mechanical Properties of the Bricks

The remediation of the surface cracks of the ancient bricks involved the configuration and curing of certain sodic particles; the process of bio-calcification during MICP could form CaCO_3_ crystals, which could, in turn, be used to fill and bind the generated gaps of the deteriorated masonry [18,19]. In our study, we show that MICP can efficiently remediate both the natural and simulated cracks in the ancient bricks. The mechanism comprises three main processes. Initially, calcium ions are adsorbed by the surface bacterial cells which are always covered by numerous negatively charged functional groups [7]. Urea is hydrolyzed to ammonium and carbonate by urease, which stands as an enriched culture medium for *Sporosarcina pasteurii* when such or similar cells are present [20,21]. Then, carbonate reacts with calcium ions and calcium carbonate crystal deposits on the cellular wall of the microorganisms [22]. Finally, the surface of the microorganisms and their surroundings are gradually wrapped as the crystal nucleus grows [23]. With the transmission of nutrients becoming restricted by the wrapped crystals, the microbes decay, and the cracks of the bricks are filled by a combination of decayed biomass and crystals [24].

Our study showed that MICP treatment increased both maximum failure load as well as the compressive strength of the cracked brick cubes, rendering their corresponding values comparable with those of the brick cubes without cracks. An increase in compression strength by MICP treatment has been also reported in cases of concrete crack remediation [25], ash bricks improvement [9], and cement mortar [26]. In addition, the wider variance of failure load and compressive strength in MICP treated sample suggested that the effect of restoration will be influenced by other uncertain factors (e.g., internal pores of the bricks [27]) that may further promote deterioration; this issue is limiting for MICP to address. Although our study may have shown that MICP delivers variance for both failure load and compressive strength after treatment, the method significantly increased the mechanical properties of the simulated cracked bricks. This indicates that MICP remediation has a great potential for the improvement and restoration of ancient bricked walls.

### 4.2. MICP Improved the Water-Resistance Property of Bricks

Pores and cracks in masonry are key factors dictating water-absorbing and/or water-resistance properties of the materials involved. Unlike pavement materials that require high water permeability, bricks used in masonry need to be adequately water resistant [28]. The pores formed during brick manufacturing and the cracks formed over time increased water absorption and decreased water resistance. With a softening coefficient > 0.8, masonry material is considered effectively water resistant [16,29]. In our study, cracks on the bricks of the Nanjing ancient city wall decreased the softening coefficient to a certain extent. This indicates that as expected cracks adversely affected the bricks in terms of water absorbance. MICP treatment could increase this coefficient to 0.93, which was significantly higher than that of the original brick cubes. The water absorption ratio experiment in this study also demonstrated that MICP can significantly reduce the bricks’ water absorption, even after 24 h immersion. Microscopy showed that MICP treatment filled the surface pores and formed an additional external layer on the surface of brick cubes. *S. pasteurii* decomposed the cementitious solution to produce calcium carbonate precipitates that were continuously deposited in the internal pores of the tested brick; the precipitate adheres the pores to the internal layers of the test brick cubes resulting in a gradual filling of the pores with calcium carbonate. Subsequently, pore volume decreased while the density of the brick cubes increased (as also described by Sun et al. [28]). In the case of water saturation, MICP-treated bricks could prevent potential damages from the internal, plausibly capillary, water. Practically, this means that an MICP-treated/enhanced wall could cope with prolonged heavy rains with minimal risk of failure. MICP has already been introduced in the concrete industry and proved its potential by promoting filling via the microbial activity of *Bacillus subtilis* [30]. Limiting cracks can assist in mitigating the development and propagation of structural failure, in concrete and not only, improving strength and structural durability [31]. Previous research reported that *Myxococcus xanthus*-induced calcium carbonate precipitation can efficiently reduce porosity in ornamental limestone [32]. The newly formed organic–inorganic composites are more stress resistant than the calcite grains of the original stone as they are strongly attached to the substratum, mostly due to the epitaxial growth of preexisting calcite grains. Our MICP-based results are in agreement with those of a similar study where MICP was applied on monumental stones [30]. As such, the efficient restoration of surface cracks via MICP would prevent crack propagation and further deterioration of brick-based ancient structures. On one hand, bacterially originated extracellular polymeric substances can form a hydrophobic protective film on the surface of the structural material (brick in this case) and form CaCO_3_, preventing water ingress. On the other hand, as the catalytic and biological reaction center is the decayed microorganisms themselves, this could form microfibers between material microparticles due to the peptidoglycan in their cell walls [24]. This potential component could plausibly be confirmed by XRF results, which showed that MICP filler holds more ‘other element’ ratio than that of the original ancient wall bricks.

Moreover, the pores near the surface are more effectively sealed by MICP than the deeper pores in the brick cubes as revealed by X-ray micro-CT. Chen et al. [33] and Sun et al. [34] also demonstrated that the internal concrete cracks could not be repaired as well as the cracks near the surface. One possible reason could be that the fluidity of the MICP medium is relatively low likely due to the silica powder mixed into the slurry [10]. MICP with pure liquid media can directly repair cracks with a width less than 0.4 mm, while for wider cracks, it is necessary to use fine sand, silica powder, and/or other fillers [35]. Another reason is that MICP is a biologically enzymic catalytic reaction, which is supposed to have a faster rate than traditional physical or chemical approaches, resulting in rapid solidification of the filling material. The surface cementation would hinder the infiltration of subsequent packing media into the deeper pores, and thus, MICP is challenged by deep pores. Hence, for deeper pores, the exploration of other technological processes and methods is necessary.

### 4.3. Material Chemistry and Mineralogy after MICP Treatment

XRD analysis showed that CaCO_3_ in calcite form was present in both ancient bricks and MICP filler. On one hand, this result validated that *S. pasteurii* can successfully utilize chemical components to produce CaCO_3_ in the brick cracks; this suggests that *S. pasteurii* has a similar function to that of *Bacillus subtilis* species as presented in former MICP-related studies [19,30]. On the other hand, the generated CaCO_3_ by MICP appears in calcite form, which is the same form as that of the ancient bricks in this study. This indicates that MICP can have better compatibility with ancient bricks than that of the commonly used materials for the reinforcement of structures (lime, cement, and epoxy) [2,6]. However, the main disadvantage of the MICP method is that the filler produced by MICP mainly consists of CaCO_3_, which could not significantly improve the acid resistance of the repaired materials. In natural environments, rain and carbon dioxide will generate water-soluble carbonic acid and/or conditions of low pH that may susceptibly erode MICP filler. In this study, we mixed silica during the MICP processing to increase acid resistance [36]. XRF showed an amorphous phase in MICP filler, which is supposed to be the purposely added silica. Previous studies have demonstrated that silica in the amorphous phase holds a higher specific surface area than that of crystal quartz, the former can more easily be combined with MICP product and bricks [37]. Ancient bricks were also found rich in silica as per the XRF analysis. Therefore, silica could be a potential amendment in MICP reinforcement technology.

## 5. Conclusions

The application of MICP for the restoration of cracked bricks from the ancient Nanjing City Wall and the evaluation of their structural properties showed that: (i) MICP is a method effective in cracks’ filling, (ii) it improves the maximum failure load of the materials, as well as (iii) their compression strength, and (iv) their water resistance. The study also highlights that the X-ray micro-CT technique can become a valuable tool for the investigation and assessment of the internal surface of bricks with multi-modal pore size distributions; this is an approach that may find wide applicability, especially in ancient masonry for the assessment of restoration at a micrometric scale. The MICP-treatment trial was performed in controlled, lab-based conditions. This leaves a degree of uncertainty on the long-term efficiency of the method for relics and their corresponding cracks when exposed to actual environments where combined adverse phenomena (temperature fluctuations, humidity, and acidic atmosphere) act in parallel. This study proves the concept of MICP-treatment for the preservation of relics; nonetheless, this method will require further optimization against actual ambient conditions.

## Figures and Tables

**Figure 1 materials-14-05665-f001:**
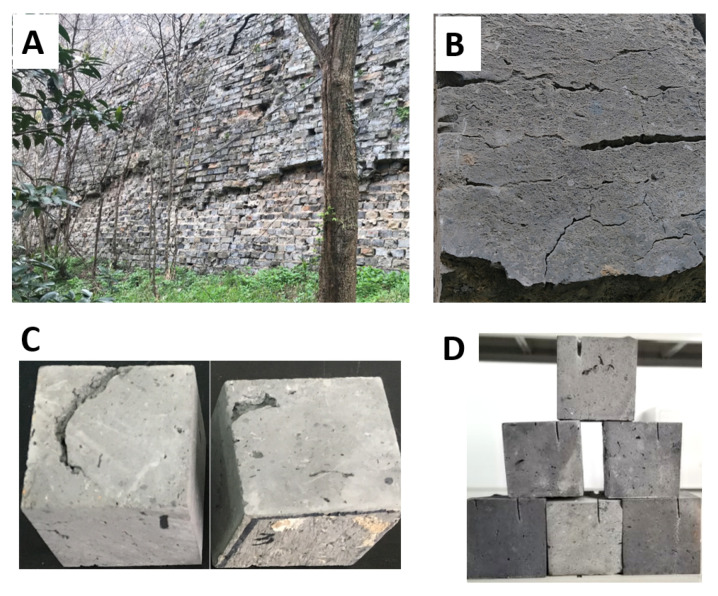
View of the city wall brick samples; (**A**) ancient Nanjing City Wall; (**B**) feature photo of wall brick; (**C**) brick cubes with natural cracks; (**D**) brick cubes with simulated cracks.

**Figure 2 materials-14-05665-f002:**
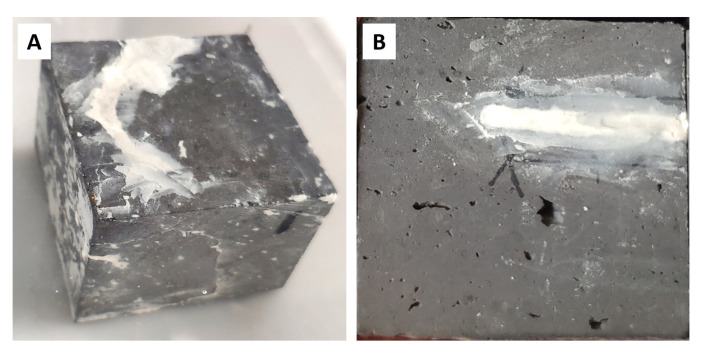
Close-up of the brick cubes after 7 days from MICP remediation. (**A**) Brick cubes with natural cracks; (**B**) brick cubes with simulated cracks.

**Figure 3 materials-14-05665-f003:**
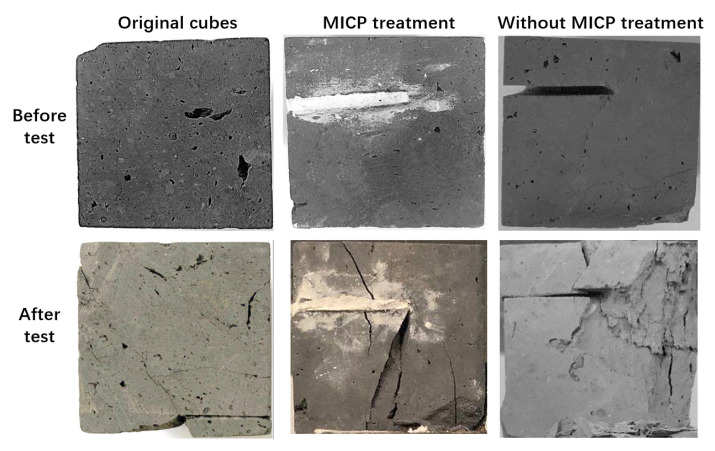
Cubes before and after compression strength test. The three photos at the top display the cubes prior test, the three at the bottom are the cubes after the test.

**Figure 4 materials-14-05665-f004:**
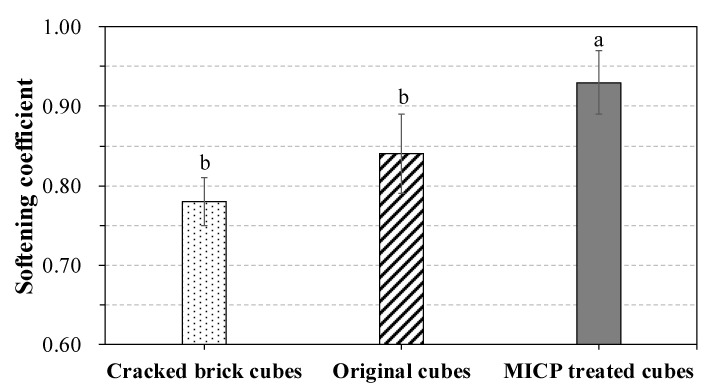
Softening coefficient for brick cubes with MICP treatment. Each measurement represents the mean of three replicate samples and bars indicate standard deviation. Different letters above the bars represent significant difference at the 0.05 level.

**Figure 5 materials-14-05665-f005:**
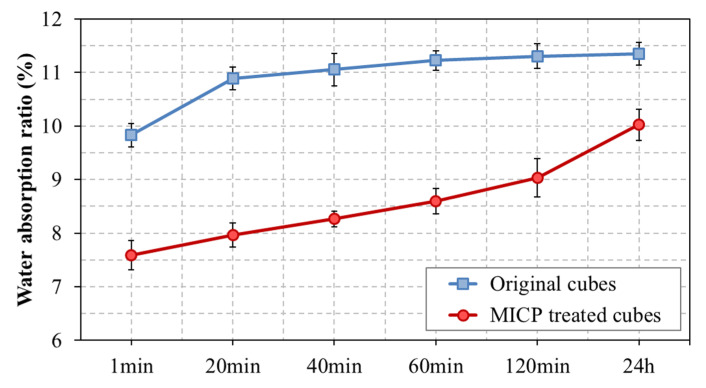
Influence of immersion time on water absorption. Each measurement represents the mean of three replicate samples and bars indicate standard deviation.

**Figure 6 materials-14-05665-f006:**
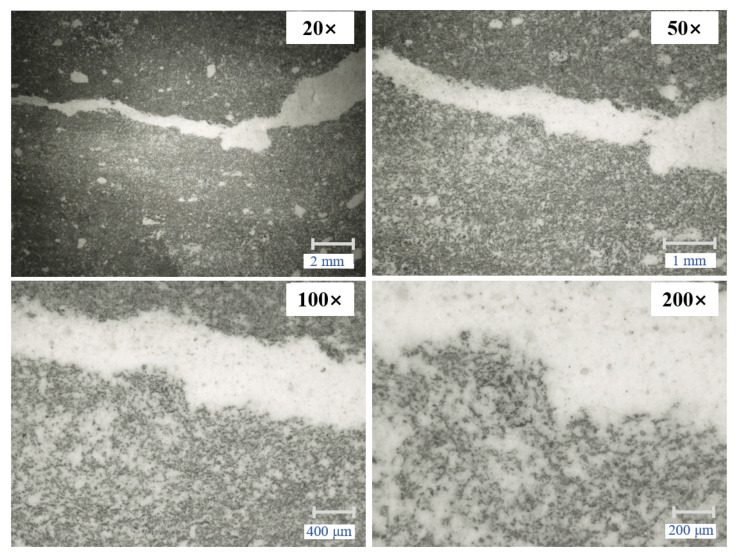
The surface of a brick cube with natural cracks restored by MICP at magnifications of 20×, 50×, 100×, and 200×.

**Figure 7 materials-14-05665-f007:**
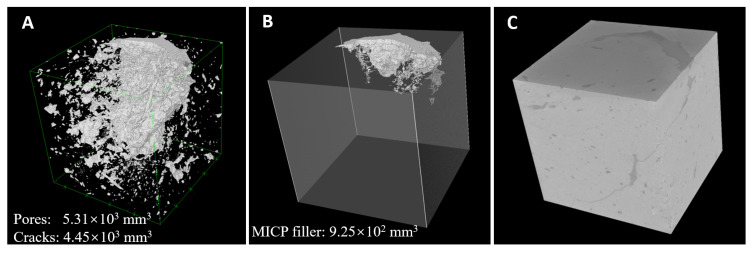
3D visualization of the brick cubes with natural cracks after MICP treatment. (**A**) Distribution of pores; (**B**) MICP filler; (**C**) the external surfaces of the MICP treated brick cube.

**Figure 8 materials-14-05665-f008:**
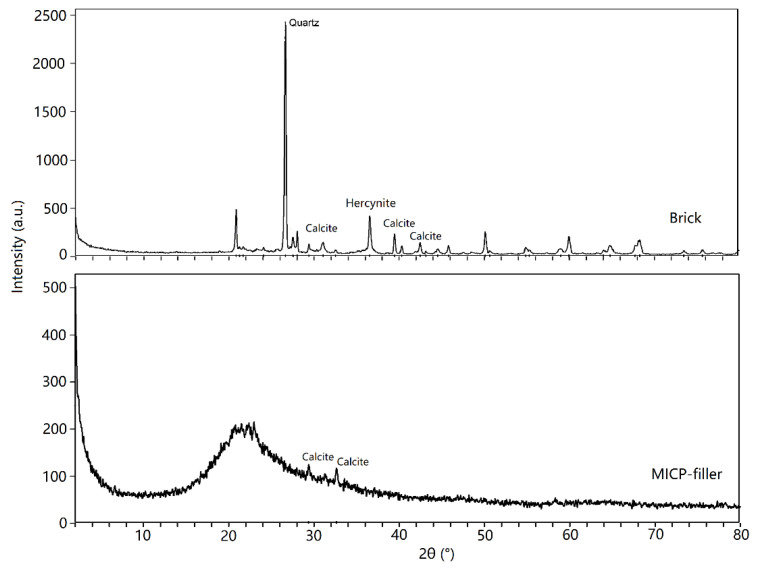
XRD results of the ancient bricks and MICP filler.

**Table 1 materials-14-05665-t001:** Advantages and Disadvantages of Commonly Used Materials for the Reinforcement of Masonry Structures.

Materials	Advantages	Disadvantages	Reference
Lime	Low cost Good compatibility	Long hardening period Low strength	[3]
Cement	Low cost Easy acquisition	Easily effloresced Inconsistent sand content Different stiffness and strength	[4]
Epoxy	Good mechanical properties High groutability	Disparate stress and strain properties Divergent thermal physical properties Low permeability and ventilation Poor interfacial properties Environmental pollution	[2]

**Table 2 materials-14-05665-t002:** Failure load and compression strength for the MICP-treated and untreated cubes with simulated cracks.

Treatments	Failure Load (kN)	Compressive Strength (MPa)
Original cubes	40.02 ± 3.84 a	20.67 ± 1.98 a
MICP remediation	33.56 ± 9.07 a	17.33 ± 4.69 a
No MICP remediation	19.00 ± 1.98 b	9.84 ± 1.02 b

Data represent the means from three replicates with standard deviation. Different letters in the column denote the significance of the differences among treatments (one-way ANOVA, *p* < 0.05).

**Table 3 materials-14-05665-t003:** Results of the major elements (wt%) detected by XRF.

Materials	SiO_2_	CaO	Al_2_O_3_	Fe_2_O_3_	P_2_O_5_	SO_3_	Cl	K_2_O	TiO_2_	Others
Brick	59.05	3.25	16.56	5.81	0.01	0.02	0.01	2.40	0.52	12.37
MICP filler	72.66	4.31	0.05	0.07	0.37	0.09	1.08	0.02	–	21.35

## Data Availability

Not applicable.

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
