# Peer review of "Microbial-Induced Carbonate Precipitation Improves Physical and Structural Properties of Nanjing Ancient City Walls"

_materials, 2021, doi:10.3390/ma14195665_

Round 1

Reviewer 1 Report

The manuscript materials-1383291, Microbial induced carbonate precipitation improves physical and structural properties of Nanjing ancient city walls, investigates an interesting restoration technique for a very ancient important Chinese monument. The experimental approach is definitely of interest with a huge readers’ potential interest, when considering the large number of ancient historical vestiges to be restored, all around the world.

However, some additional data and pieces of information are necessary for assessing the interest of this method vs. existing alternatives. For instance:

  1. In the introductory part, please make a brief overview of the restoring techniques for such type of monuments (materials), with their advantages vs limits. Could be in a Table format, with the corresponding references in a separate column

  1. Please present a general climatic image of the Nanjing region (winter and summer temperatures, temperature gap day/night, mean humidity and general annual evolution, existence or not of industrial/transport pollution nearby, acid rains or not and so on). These pieces of information are necessary to conceive realistic accelerated aging tests, in order to assess on the pertinence of using MICP method for restoring Nanjing ancient city walls.

  1. Several questions appeared, after reading the experimental part:

  • Could authors assess on cracks behavior during the cutting step with the electric saw? Possibly cracks propagation?
  • Could authors assess on the typical size and shape of cracks, as well as on their location (on surface vs internal, communicating pores vs closed pores), before and after MICP restoration? These experimental data are needed to better understand the mechanical values obtained. How many samples have been tested, for each type of cubes?
  • The cubes with natural cracks used in this work were having all (or in great majority) the same kind of cracks? And which kind exactly? Which pore width and depth, in average and the extreme values? Mainly connected pores or in surface? What about the type of simulated cracks? Where they of similar nature /size / shape / volume density and location as the natural cracks? Could you present pore maps as obtained by X-ray micro-CT (a very pertinent 3D noninvasive method)?
  • In section §3.2, Figure 3, I suggest magnifying the “after test” images with focus on the cracked cubes, rather than on the testing cell. Could you please discuss on the type of cracks after the compression test? For instance – were they in majority new cracks on untreated zones of the cubes or around the treated cracks and so on? These valuable experiments deserve a more complete assessment and discussion, and also a critical discussion vs. literature data for other types of restoring methods.
  • The discussion in section §4.2 is a very good example of critical comparison with literature data. Please remind here if Nanjing region has typical harsh winter conditions? Could the infiltrated water follow repetitive freeze/thaw cycles during winter and thus widen the pores?
  • Could authors propose some potential ideas for improving the MICP restoration technique? Some potential additives to be added inside the chosen MICP formulation?
  • Accelerated ageing tests (combining temperature, humidity and controlled acid atmosphere) could offer valuable pieces of information about the stability of the treated cracks vs. their natural neighbourhood, and the general resistance in time of the treated parts and the overall treated cubes.

  1. In Conclusion part – please remove the sentence from lines 353-354. Also, the last paragraph (lines 361-365) is very pertinent, but this aspect deserves a larger discussion in the previous sections and some potential remediation ways, not only to be mentioned in conclusion.

To conclude, I consider this work as very interesting, but the experimental results and observations deserve a more effective qualitative and quantitative valorization of the experimental observations (as mentioned above).

I recommend minor revision of the present manuscript with improvement in all suggested points, for scientific soundness and for increasing the readers' interest.

Reviewer 2 Report

It is very interesting manuscript.

In this study, X-ray micro computed tomography (X-ray micro-CT) was employed for non-destructively visualize the internal structure of the MICP-treated brick cubes.

But there is no data of material chemistry and mineralogy of Nanjing ancient city walls as well as chemical composition and mineralogy of precipitated materials via MICP processes.

Therefore, additional data of both material chemistry/mineralogy of Nanjing ancient city walls and chemical composition/mineralogy of precipitated materials via MICP processes before and after MICP treatment. Without these data, we don’t know how MICP can effectively repair ancient relics, promoting durability and limiting degradation without affecting the structure.

In my opinion, additional data of both precipitated material chemistry (i.e., chemical composition of the precipitated materials using SEM-EDS analyses) and mineralogy (i.e., mineralogy of the precipitated materials using XRD analyses) and material chemistry (i.e., chemical composition of the Nanjing ancient city walls using SEM-EDS analyses) and mineralogy (i.e., mineralogy of tNanjing ancient city walls using XRD analyses) are essential to support mechanism of improvement   physical and structural properties of Nanjing ancient city walls after microbial induced carbonate precipitation.

Actually, we do not know what kinds of materials are precipitated on or cracks of the Nanjing ancient city walls via MICP processes. Using current data on the manuscript, we don’t know how MICP can effectively repair ancient relics, promoting durability and limiting degradation without affecting the structure. 

Round 2

Reviewer 2 Report

The authors revised the manuscript according to my review comments. The manuscript is acceptable to materials.